# Deciphering the Complex Intertwining Between Cytopenia and Transfusion Needs After CAR-T-Cell Therapy for B-Cell Malignancies

**DOI:** 10.3390/life15091419

**Published:** 2025-09-09

**Authors:** Claudio Pellegrino, Eugenio Galli, Patrizia Chiusolo, Rossella Ladiana, Caterina Giovanna Valentini, Marcello Viscovo, Federica Sorà, Simona Sica, Luciana Teofili

**Affiliations:** 1Dipartimento di Scienze di Laboratorio ed Ematologiche, Fondazione Policlinico Universitario “A. Gemelli” IRCCS, 00168 Rome, Italy; eugenio.galli@policlinicogemelli.it (E.G.); patrizia.chiusolo@unicatt.it (P.C.); rossella.ladiana1@guest.policlinicogemelli.it (R.L.); caterinagiovanna.valentini@policlinicogemelli.it (C.G.V.); marcello.viscovo@guest.policlinicogemelli.it (M.V.); federica.sora@unicatt.it (F.S.); simona.sica@unicatt.it (S.S.); luciana.teofili@unicatt.it (L.T.); 2Sezione di Ematologia, Dipartimento di Scienze Radiologiche ed Ematologiche, Università Cattolica del Sacro Cuore, 00168 Rome, Italy

**Keywords:** CAR-T, cytopenia, transfusion

## Abstract

Immune-effector-cell-associated hematotoxicity has emerged as the most common CAR-T-cell-related complication in the real-world setting. Therefore, transfusion of blood components remains unavoidable in many patients treated with CAR-T cells to alleviate symptomatic anemia and prevent major bleeding events. This study investigates predictive factors associated with the transfusion requirement in patients receiving anti-CD19 CAR-T-cell therapy for B-cell malignancies in a real-world setting and the potential correlation between transfusion needs, ICAHT, and long-term survival outcomes. Among 90 investigated patients, 51 (56.7%) received at least one transfusion in the three months post-infusion (33.4% received only RBC concentrates, and 23.4% received both RBC and platelet transfusions). The highest transfusion needs occurred in the first month post-infusion, with 50 transfused patients (55.5%). Early transfusion-requiring cytopenia was associated with pre-infusion altered bone marrow function, patients-related factors, including female sex, and acute inflammatory toxicities. The incidence of late cytopenia was mainly predicted by the need for pre-infusion transfusion support. Patients receiving platelet transfusions were characterized by an inferior progression-free (*p* = 0.013) and overall survival (*p* = 0.005). CAR-T-cell-treated patients can experience a high transfusion burden, impairing their quality of life, potentially affecting survival outcomes, and resulting in overutilization of clinical resources

## 1. Introduction

Immune-effector-cell-associated hematotoxicity (ICAHT) represents the most frequently observed adverse event linked to CAR-T-cell therapy, both in clinical trials and real-world practice [1,2].

Overall, severe (grade ≥ 3) neutropenia, thrombocytopenia, and anemia have been reported in different ranges (29% to 95%, 15% to 65%, and 14% to 77%, respectively) for commercial anti-CD19 products [3].

CAR-T HEMATOTOX has emerged as the most adopted model for predicting severe CAR-T hematotoxicity in adult patients with R/R B-cell malignancies. Pre-infusion bone marrow reserve and baseline inflammatory markers are associated with a longer duration of neutropenia, a higher incidence of severe thrombocytopenia and anemia, infectious complications, and poorer survival outcomes [4,5].

More recently, an innovative grading system has been developed by EBMT and EHA, based on the depth and duration of neutropenia and used for scoring early (day 0–30) and late (after day 30) cytopenia [5,6]. Severe ICAHT is associated with a higher rate of severe infections, increased non-relapse mortality, and inferior survival outcomes [7].

Current guidelines recommend red blood cell (RBC) and platelet (PLT) transfusions after CAR-T-cell therapy as a supportive measure for patients experiencing severe cytopenia [6]; accordingly, up to 55–66% of patients after CAR-T-cell infusion receive at least one transfusion [8]. Nonetheless, the interplay between transfusion support and outcomes remains poorly characterized.

For red blood cell transfusion, a hemoglobin threshold of 7–8 g/dL is accepted for hemodynamically stable patients, while platelet concentrate transfusions are indicated for patients with platelet counts ≤ 10 × 10^9^ /L or for those with active bleeding, fever, or ongoing infections, for whom the threshold is raised to ≤20 × 10^9^ /L [8]. These transfusion thresholds are largely based on evidence from the autologous stem cell transplantation literature, as there is a paucity of specific data for the setting of CAR-T-cell therapy. Moreover, it is imperative that red blood cells and platelets be irradiated, in part because of the prior exposure to fludarabine, which can increase the risk of transfusion-associated graft-versus-host disease [8].

This monocentric retrospective observational study aims to identify predictive factors associated with the transfusion requirement in patients receiving anti-CD19 CAR-T-cell therapy for B-cell malignancies in a real-world setting. Secondly, it examines the potential correlation between transfusion needs, hematological toxicity, and long-term survival outcomes.

## 2. Materials and Methods

Consecutive patients treated with commercial anti-CD19 CAR-T cells at Fondazione Policlinico Gemelli IRCCS between September 2019 and June 2024, with a minimum follow-up of 3 months post-infusion, were retrospectively included in the analysis. Anonymized data were retrieved from the institutional database and encompassed patient demographics, disease features, hematological history (including the number and types of prior therapies and bridging treatments), baseline bone marrow assessments, the lymphodepletion regimen, early CAR-T-related toxicities (CRS, ICANS), CAR-T expansion kinetics [9], and transfusion support. A bone marrow examination was performed in all cases per routine center practice to evaluate disease infiltration and baseline medullary cellularity. Hypocellularity was defined by reference to age-specific normal ranges of hematopoietic area, as adapted from the literature [10].

Grading of CRS and ICANS obtained by clinical records had been prospectively assessed and documented by the treating teams following the American Society for Transplantation and Cellular Therapy (ASTCT) consensus criteria [11]. Toxicity management adhered to international guidelines [12].

The CAR-HEMATOTOX score was calculated by the investigators using laboratory values collected on the day of lymphodepletion initiation (day 5) ±3 days [3]. Hematological toxicity was assessed according to the Common Terminology Criteria for Adverse Events (CTCAE) version 5.0 scoring system and the EHA/EBMT grading system for ICAHT [6]. The clinical phenotype of neutrophil recovery was additionally classified as intermittent recovery (neutrophil recovery > 1500 cells/µL followed by a second dip with an absolute neutrophil count < 1000 cells/µL after day 21), quick recovery (sustained neutrophil recovery without a second dip below <1000 cells/µL), and aplastic (continuous severe) neutropenia (absolute neutrophil count < 500 cells per µL for ≥14 days) [4].

Institutional guidelines for red blood cell and platelet transfusions remained unchanged throughout the study period. Transfusion thresholds were aimed at maintaining a hemoglobin level > 8 g/dL and a platelet count > 10,000/μL or 20,000/μL in patients without or with hemorrhagic syndrome (WHO grade ≥ 2) [13], respectively. Each pool consisted of a leukocyte-depleted, pathogen-inactivated platelet component derived from 4 to 6 fresh whole-blood donations. Both pooled platelet products and apheresis platelets contained a minimum of 2 × 10^11^ platelets per unit. All administered blood components were irradiated [14].

Transfusion requirements were defined as the number of RBC concentrates and PLT units, either as apheresis or pool PLT products, received in the first 3 months after CAR-T infusion.

Transfusion support was dichotomized as early support (day 0–30) or late support (day 31–90). A high transfusion burden was defined as receiving ≥5 blood products post-infusion.

A Sankey diagram was generated to visualize transitions in transfusion status over time. The diagram represents the number of patients receiving red blood cells or platelets during specific time intervals (the three months before and after CAR-T infusion). The width of the flows is proportional to the number of patients moving between transfusion categories.

Transfusion and relapse-free survival (TRFS) was defined as survival from CAR-T-cell infusion and the last follow-up without transfusions or relapse, new treatment onset, or death.

Responses to CAR-T-cell therapy were assessed with 18FDG PET–CT scans and evaluated based on the Deauville/Lugano criteria for LBCL and MCL and by bone marrow MRD assessment for B-ALL [15].

Overall survival (OS) was defined as the time from CAR-T-cell infusion and the last follow-up or death from any cause. Progression-free survival (PFS) was calculated as the interval between CAR-T-cell infusion and the documentation of disease persistence (imaging or histology) or death from any cause.

### Statistical Analysis

Continuous variables are expressed as the median and IQR and categorical variables as absolute and relative frequencies. All continuous variables were analyzed for normality with the Shapiro–Wilk test. To compare continuous variables, we used the Mann–Whitney or Student’s *t*-test as appropriate; for categorical variables, we used Fisher’s exact test or the χ^2^ test as appropriate. The combined effect of different variables was evaluated by multivariate logistic regression analysis (using the backward stepwise method) after centering collinear variables and including as covariates those with a significant effect (*p* < 0.05) on the outcome at univariate analysis.

Survival analyses were performed using the Kaplan–Meier methodology. Groups were compared by the log-rank test (OS, PFS) or the Fine and Grey method (TFRS) and are expressed as a hazard ratio (HR) with relative 95% confidence intervals (CIs). Cox proportional hazard regression models were applied to estimate hazard ratios with a 95% CI, adjusting for potential confounders.

Tests were performed with NCSS 2020 Statistical Software (2020) LLC. Kaysville, UT, USA and GraphPad Prism v 8.0.

## 3. Results

A total of 90 consecutive patients were included in this study. Their clinical data are summarized in Table 1.

The CAR-T-cell therapy indication was large B-cell lymphoma in 72 patients (80.0%), mantle cell lymphoma in 11 patients (12.2%), and B-cell acute lymphoblastic leukemia in 7 patients (7.8%). Overall, 49 (54.4%) received axi-cel, 24 (26.7%) tisa-cel, and 17 (18.9%) brexu-cel. All patients received fludarabine-cyclophosphamide-based lymphodepletion according to the manufacturer’s instructions.

At the time of CAR-T-cell infusion, the median age was 57 years (IQR: 47.0–65.0), and the median Eastern cooperative oncology group (ECOG) performance status was 1 (IQR 0–1). The median number of prior therapy lines was 2 (IQR 2–3). Eighty-four patients received bridging therapy, consisting of systemic chemoimmunotherapy in thirty-three patients (36.7% of cases), local radiotherapy in twenty-six patients (28.9%), and autologous stem cell transplantation in nine patients (10.0%). Bridging resulted in disease control (CR or PR) in 38 of 90 patients (42.2%).

Before starting lymphodepletion, 48 patients (53.3%) presented with reduced bone marrow cellularity, while 12 (13.3%) showed bone marrow disease infiltration. The CAR-HEMATOTOX score was high (≥2) in 50 (60.0%) patients.

CRS occurred in 90.0% of cases (13 (14.5%) patients developed a CRS ≥3, while the median of tocilizumab doses/patient was 1 (IQR 0–3)). Thirty patients (33.3%) developed signs of ICANS (in eleven patients (12.2%), the ICANS grade was ≥3). The clinical course was complicated by severe (>2) early ICAHT in 26 patients (28.8%) and by severe (>2) late ICAHT in 25 patients (27.7%).

At a median follow-up of 10.8 months (IQR 5.1–23.0), the mortality rate for the entire cohort was 30.0% (*n* = 27). Twenty-two patients died of primary disease progression or relapse (disease-related mortality). Five patients (4.4%) experienced non-relapse mortality.

The best overall response rate (ORR) was 69.7%, including 47.6% of patients who achieved a complete response (CR) and 22.7% with a partial response (PR).

Median OS and PFS were 40.0 months (95% CI 23.7—not reached) and 16.8 months (95% CI 6.6–28.6), respectively.

### 3.1. Transfusion Requirements

Among the 90 investigated patients, 33 (36.7%) had received at least one transfusion in the 3 months preceding the CAR-T-cell infusion (10 patients (9.0%) had received only RBCs, 4 patients (4.4%) had received only platelets, and 19 patients (21.1%) had received both blood products. The median number of transfused RBC units was 0 per patient (range, 0–16), while the median number of platelet units was 0 per patient (range, 0–12).

After CAR-T-cell infusion, 51 patients (56.7%) were transfused (30 patients (33.4%) were given only RBCs, and 21 (23.4%) were given both RBC and platelet transfusions, Figure 1). The median units of transfused RBCs and PLTs were 1 (range: 0–18) and 0 (range: 0–20) per patient, respectively.

When considering only patients requiring any type of transfusion, the average need for transfusions during the first three months after CAR-T was 2/pts (IQR 1–5) for RBCs and 1/pts (IQR 0–4) for PLTs.

Patients requiring transfusions after CAR-T-cell therapy had a similar age, ECOG PS, hematological diagnosis, number of prior lines of treatment, type of bridging therapy, and CAR-T product infused compared to non-transfused patients (Table 1). Transfused patients were more frequently female (*p* = 0.001) and presented with a higher CAR-HEMATOTOX score (*p* < 0.001) and reduced bone marrow cellularity (*p* < 0.001). These patients also exhibited higher CRS incidence (0.033), higher CRS severity (*p* = 0.008), and a trend toward higher circulating IL-6 peak levels (*p* = 0.054). Moreover, severe early and late ICAHT incidence was greater in the transfused group (*p*= 0.005 and *p* = 0.001, respectively).

There was no significant difference in CAR-T-cell expansion in peripheral blood when evaluated at +7, +14, and +30 days after infusion.

The monthly need for transfusions immediately before and after CAR-T infusion is summarized in Figure 2.

The highest transfusion needs occurred in the early phase (within the first month), with 50 patients (55.5%) requiring at least one transfusion (48 patients (53.4% were transfused with RBC units (mean number 1.3/patient, range: 0–11) and 16 (17.8%) with platelet units (mean number 0.6/patient, range: 0–12)).

Subsequently, the number of transfused patients decreased to a minimum of 12 in the third month (13.4%). In parallel, the mean number of transfused units decreased to 0.3/patient (range: 0–6) for RBCs and 0.2/patient for platelets (range: 0–7). No CAR-T patient developed alloantibodies against RBC antigens following transfusions.

Fifteen patients (16.6%) received support with Erythropoietin (EPO), fourteen of which were in the RBC-transfused group (*p* < 0.001). Three patients in the transfused group also received a hematopoietic stem cell boost due to prolonged cytopenia (autologous *n* = 2; allogeneic *n* = 1), leading to complete hematological recovery.

### 3.2. Predictive Factors of Post-CAR-T-Cell Transfusion Needs

Transfused and non-transfused groups were compared for factors potentially associated with transfusion requirements (Appendix A). Clinically relevant and statistically significant variables were then included as covariates in a multivariate logistic regression model (Table 2).

Factors significantly associated with the need for early RBC transfusion were receiving at least one transfusion before CAR-T-cell therapy (aOR 8.7, 95% CI 2.1–36.9, *p* = 0.002), a high (≥2) CAR-HEMATOTOX score (aOR 5.1, 95% CI 1.4–18.7, *p* = 0.013), female sex (aOR 6.3, 95% CI 1.7–23.3, *p* = 0.005), and no response to bridging therapy (aOR 4.4, 95% CI 1.2–16.0, *p* = 0.026).

The only factor significantly associated with the need for late RBC transfusion was receiving at least one transfusion before CAR-T-cell infusion (aOR 7.6, 95% CI 1.4–41.8, *p* = 0.019).

Factors significantly associated with the need for early platelet transfusion included ECOG PS ≥ 2 at infusion (aOR 4.9, 95% CI 1.1–22.5, *p* = 0.041), having received at least one transfusion before CAR-T therapy (aOR 8.3, 95% CI 1.7–41.3, *p* = 0.009), and severe (grade > 2) ICANS (aOR 6.3, 95% CI 1.1–38.6, *p* = 0.042).

Factors significantly associated with the need for late platelet transfusion were severe (>2) early ICAHT (aOR 4.9, 95% CI 1.1–22.2, *p* = 0.036) and having received at least one transfusion before CAR-T-cell infusion (aOR 8.2, 95% CI 1.5–46.1, *p* = 0.017).

### 3.3. Correlation Between Transfusion Support and Outcomes

The overall response rate was not statistically different between transfused and non-transfused patients (*p* = 0.073, Table 1). In contrast, when differentiating by transfusion timing, only late platelet transfusion was associated with a lower ORR (*p* = 0.016).

TFRS negatively correlated with disease response (*p* = 0.041) but showed no association with disease progression (*p* = 0.376) or death (*p* = 0.316) (Appendix A).

Nevertheless, shorter TFRS correlated with a higher total transfusion burden (HR 4.56, 95% CI 1.03–20.20, *p* < 0.001) and was more frequently associated with neutropenia, classified as early (HR 2.22, 95% CI 1.02–5.02, *p* < 0.001) and late ICAHT (HR 2.61, 95% CI 1.17–5.80, *p* < 0.001) (Appendix A).

Neither PFS nor OS were affected by post-CAR-T cell transfusion of RBCs (*p* = 0.175 and *p* = 0.212, respectively) (Figure 3A,B, Table 3). In contrast, patients receiving any platelet transfusions had significantly lower PFS (HR 2.14, 95% CI 1.11–4.50, *p* = 0.013) and OS (HR 3.12, 95% CI 1.17–8.33, *p* = 0.001) (Figure 3C,D, Table 3). This association was predominantly determined by early platelet transfusion (HR 2.19, 95% CI 1.05–5.04, *p* = 0.016 and HR 2.73, 95% CI 1.11–8.29, *p* = 0.011 for PFS and OS, respectively) (Table 3; Appendix A). Conversely, late platelet transfusion showed a trend toward worse overall survival (*p* = 0.056) but did not significantly affect progression-free survival (*p* = 0.421) (Table 3; Appendix A).

In multivariable Cox regression analysis adjusted for age, number of previous lines of therapy, disease state at infusion, ECOG PS, bone marrow cellularity, severe early ICAHT and severe late ICAHT, receiving any platelet transfusion remained significantly associated with PFS (HR 3.62, 95% CI 1.43–9.14, *p* = 0.006) and OS (HR 7.17, 95% CI 2.21–23.23, *p* = 0.001).

## 4. Discussion

The introduction of immunotherapy with genetically engineered chimeric antigen receptor T-cells has rapidly revolutionized the therapeutic landscape of B-cell malignancies [16]. However, this unparalleled efficacy has come at the cost of a unique toxicity profile. Although CRS and ICANS represent the prototypical adverse events associated with CAR-T-cell therapy, ICAHT has emerged as one of the most common CAR-T-cell-related complications in the real-world setting [2]. The pathophysiology of ICAHT remains only partially elucidated. Cytopenia probably results from the intertwining of multiple factors, including altered bone marrow function pre-lymphodepletion, the toxicity of the lymphodepleting chemotherapy, the immune response elicited by CAR-T-cells upon antigen recognition and subsequent expansion, and transient factors such as infections [2].

Various therapeutic strategies have been proposed, including growth factors administration (EPO, G-CSF, and TPO-RAs) and hematopoietic stem cell boosts [2]. Despite these efforts, transfusion of blood components remains unavoidable in a large fraction of CAR-T-cell-treated patients to alleviate symptomatic anemia and prevent major bleeding events.

This population could represent a distinct subset of severely cytopenic individuals, and their characteristics and outcomes warrant further investigation. Additionally, it has been suggested that the immunomodulation triggered by blood product administration could influence the function of cellular therapies and ultimately impact prognosis [8].

In this context, transfusion-related immune modulation (TRIM) refers to a complex set of responses triggered by allogeneic blood products, involving both immunosuppressive and pro-inflammatory effects, driven by residual leukocytes, apoptotic cells, microparticles, and soluble factors [17]. TRIM involves suppression of cytotoxic and monocyte activity, increased regulatory T-cells, and release of immunosuppressive mediators [17]. Such immunomodulatory effects, which may partly explain associations between perioperative allogeneic transfusions and post-surgical cancer recurrence [18], could also subtly influence CAR-T-cell function and persistence, although their impact in this setting remains poorly defined.

In our cohort, approximately two-thirds of patients received at least one transfusion within the first three months following CAR-T-cell therapy, with the highest percentage occurring in the first month for both RBCs and PLT concentrates. Notably, the average number of transfused RBCs/patient was comparable to that observed in the autologous hematopoietic stem cell transplantation setting [19]. These figures highlight how a CAR-T program could pose a significant burden for blood services.

Next, we analyzed data on longitudinal neutrophil kinetics to assess our cohort using the EHA/EBMT ICAHT grading system. Our findings revealed a statistical connection between blood transfusions and early and late ICAHT development, suggesting that patients receiving transfusions exhibit a multilineage aplastic phenotype.

The mechanisms underlying early transfusion-requiring cytopenia are likely multifactorial. First, we observed that being on transfusion support prior to the infusion strongly predicted the need for both RBC and platelet transfusions, as shown by multivariate analysis. This observation aligns with previous findings [8] and may imply a preexisting impairment in bone marrow function, which can be worsened by the toxicity associated with bridging and lymphodepleting therapies. In accordance, transfused patients showed reduced bone marrow cellularity.

A high-risk CAR-HEMATOTOX score, which includes baseline inflammatory state data, proved to be an independent risk factor for early RBC administration but not for PLT. Notably, in our cohort, the platelet transfusion thresholds were lower than the cut-off used to validate the score, which may have diminished its predictive accuracy.

Female patients required more RBC transfusions in the early phase, despite having similar baseline hemoglobin and MCV compared with males. This difference was not observed for platelet use, suggesting a red-cell-specific phenomenon. Potential explanations include sex-related differences in iron stores, blood volume, erythroid recovery, or chemotherapy pharmacokinetics. This finding warrants further study, given the known influence of sex on both erythropoiesis [20] and CAR-T outcomes [21].

In line with previous observations, we demonstrated an association between a high ECOG PS and early platelet transfusions, reinforcing the connection between a poor functional status and higher treatment-related toxicity [8].

Adverse early inflammatory events have been linked to impaired hematopoietic recovery [22,23]. Indeed, severe ICANS increased the risk of early platelet transfusions. In analogy, higher levels of circulating inflammatory cytokines and severe CRS were associated with early RBC transfusion, although multivariate analysis did not confirm this finding.

Despite evidence indicating increased CAR-T-cell toxicity, including ICAHT, with the CD28 costimulatory domain (axicabtagene ciloleucel and brexucabtagene autoleucel) compared with 4-1BB–based products (tisagenlecleucel) [8,24], we did not observe a significant impact of product type on transfusion requirements. Moreover, CAR-T-cell expansion kinetics in peripheral blood did not significantly differ between transfused and non-transfused patients, even when analyses were stratified by transfused component (RBCs vs. platelets) or by time window (day 0–30 vs. day 30–90).

On the contrary, the genesis of late cytopenia remains more elusive. The strongest predictive factor was, once again, the need for pre-infusion transfusion support.

We finally evaluated the impact of transfusion on clinical outcomes, highlighting a specific role for platelet concentrates. Receiving at least one platelet transfusion predicted inferior progression-free survival and overall survival, while we demonstrated no clear prognostic role for RBCs. Similarly, lower transfusion-free survival post-infusion was associated with ICAHT development and an inferior response rate.

Although it is tempting to evocate a causal association, as transfusion may modulate the recipient’s inflammatory state and immune response [25], there are several caveats to consider. First, transfused and not-transfused patients presented some baseline differences that may be confounding factors affecting relapse and survival, which were not taken into account in the analysis. Secondly, we reported an association between transfusion requirements and ICAHT, which is known to predict inferior outcomes [7]. We partially mitigated these limitations by confirming the influence of platelet transfusions on survival outcomes in a Cox regression analysis, but additional unknown confounders cannot be excluded.

Finally, future studies may benefit from assessing the cumulative effect of the transfusion burden, as the number of transfusions may better capture long-term toxicities, such as iron overload, than a dichotomous transfusion status.

Nevertheless, our findings reinforce the notion of post-CAR-T transfusions as a negative prognostic factor and may guide preemptive interventions such as intensive anti-infective prophylaxis, early growth factor support, or storage of backup hematopoietic stem cell boosts.

Studies specifically addressing transfusion requirements in patients undergoing CAR-T therapy have so far been scarce. Vic et al. retrospectively analyzed data from 671 patients in the French DESCAR-T registry, which represents the largest published cohort with comprehensive transfusion data [8]. Compared with their study, our analysis has several limitations, including its monocentric design, the smaller sample size, and the heterogeneity of the included cohort, all of which may limit the generalizability of the results. Conversely, our study benefited from the use of well-defined and consistent transfusion thresholds throughout the entire observation period and from the integration of additional variables, such as patient sex, bone marrow status, circulating inflammatory cytokine levels, CAR-T-cell expansion kinetics, and longitudinal neutrophil counts [26]. Notably, our findings are consistent with those of Vic et al., despite being obtained in a different healthcare setting (Italy versus France) characterized by differences in the transfusion service organization and access to CAR-T therapies [27], thereby reinforcing the robustness of the results.

## 5. Conclusions

CAR-T-cell-treated patients can experience a high transfusion burden, impairing their quality of life, potentially affecting outcomes, and resulting in overutilization of clinical resources. To this end, it is crucial to expand our understanding of the pathogenesis of hematological toxicity in the context of cellular immunotherapies.

In the meantime, well-designed prospective studies are needed to thoroughly characterize the prognostic role of transfusion and optimize transfusion management.

## Figures and Tables

**Figure 1 life-15-01419-f001:**
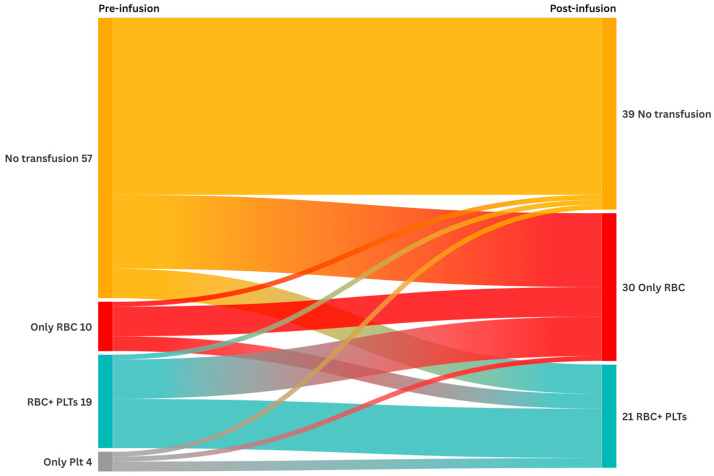
Sankey diagram representing patients in terms of transfusion support in the 3 months preceding and after CAR-T-cell infusion. For each timepoint, patients are categorized into four groups according to blood components received (No transfusion, RBCs, PLTs, or RBCs + PLTs). The majority of patients already on transfusion support before CAR-T received at least one transfusion in the 3 months post-infusion.

**Figure 2 life-15-01419-f002:**
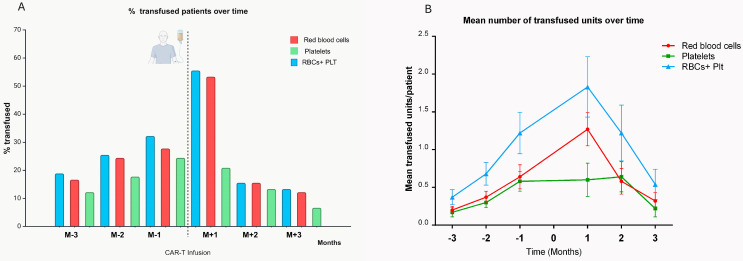
Evolution of transfusion needs in the 3 months before and after CAR-T-cell infusion. (**A**) Percentage of patients who received transfusions and (**B**) the mean number of RBC units transfused per patient, with the standard error of the mean (SEM). The highest transfusion requirements were observed during the first month following infusion, with 50 patients (55.5%) requiring at least one transfusion, predominantly of red blood cells (RBCs), either alone or in combination with platelets. Following this peak, the number of transfused patients declined, with a balanced contribution of RBC and platelet (Plt) units.

**Figure 3 life-15-01419-f003:**
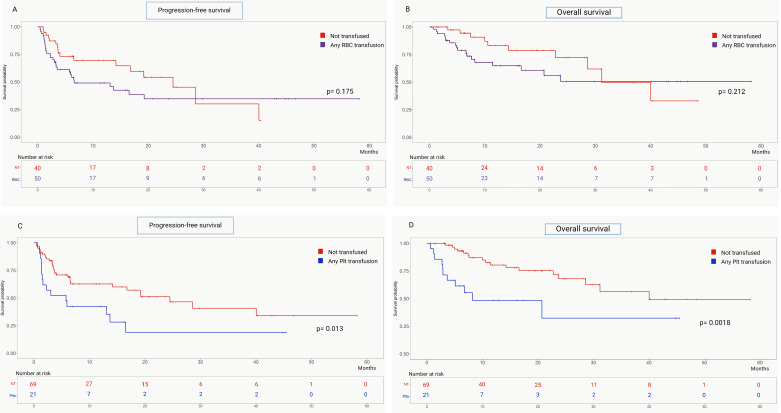
Comparison of progression-free survival (**A**) and overall survival (**B**) in patients receiving at least one RBC transfusion post-CAR-T-cell infusion and those who did not. Comparison of progression-free survival (**C**) and overall survival (**D**) between patients who received at least one Plt transfusion post-CAR-T-cell infusion and those who did not.

**Table 1 life-15-01419-t001:** Patients’ baseline characteristics, post-infusion variables for the entire cohort, and a comparison between those receiving at least one transfusion after CAR-T-cell infusion versus no transfusion. Data are given as *n* (%) for categorical variables and median (IQR) for continuous variables. Values in bold signify statistical significance (*p* < 0.05).

	Entire Cohort (n = 90)	At Least One Transfusion (n = 50)	No Transfusion (n = 40)	*p*-Value
Disease Histology (*n*, %)				
DLBCL	72 (80.0)	39 (78.0)	33 (82.5)	
MCL	11 (12.2)	8 (16.0)	3 (7.5)	0.388
B-ALL	7 (7.8)	3 (6.0)	4 (10.0)	
Sex (M/F, %)	40/50	20/30 (40.0)	30/10 (75.0)	**0.001**
Age at infusion (Years)	57 (47.0–65.0)	56.0 (46.5–63.2)	57.8 (51.0–65.0)	0.572
ECOG PS at CAR-T infusion (Score)	1 (0–1)	1 (1–2)	1 (0–1)	**0.038**
Prior lines (number)	2 (2–3)	2 (2–4)	2 (2–3)	0.383
Prior lines > 3 (*n*, %)	17 (18.9)	14 (28.0)	3 (7.5)	**0.013**
Bridging therapy (*n*, %)				
None	6 (6.7)	4 (8.0)	2 (5.0)	
ASCT	9 (10.0)	8 (16.0)	1 (2.5)	0.078
Chemoimmunotherapy	33 (36.7)	20 (40.0)	9 (32.5)	
Local radiotherapy	26 (28.9)	10 (20.0)	11 (40.0)	
Others *	16 (17.7)	8 (16.0)	6 (20.0)	
Partial/Complete response to bridging therapy (*n*, %)	38 (42.2)	15 (30.0)	23 (57.5)	**0.010**
RBCs transfused within 3 months before CAR-T-cell therapy (units)	0 (0–2)	1 (0–2)	0 (0–0)	**<0.001**
Platelets transfused within 3 months before CAR-T-cell therapy (units)	0 (0–1)	0 (0–1)	0 (0–0)	**<0.001**
CAR-HEMATOTOX score high risk (≥2) (*n*, %)	54 (60.0)	40 (80.0)	14 (35.0)	**<0.001**
BM status at the infusion (*n*, %)				
Reduced cellularity	48 (53.3)	36 (72.0)	12 (30.0)	**<0.001**
Disease infiltration	12 (13.3)	8 (16.0)	4 (10.0)	0.405
CAR-T product infused (*n*, %)				
Axi-cel	49 (54.4)	26 (52.0)	23 (57.5)	
Tisa-cel	24 (26.7)	14 (28.0)	10 (25.0)	0.872
Brexu-cel	17 (18.9)	10 (20.0)	7 (17.5)	
CRS incidence (*n*, %)	81 (90.0)	48 (96.0)	33 (82.5)	**0.033**
CRS severity (*n*, %)				
CRS grade 1	27 (30.0)	13 (26.0)	14 (35.0)	
CRS grade 2	41 (45.5)	23 (46.0)	18 (45.0)	**0.008**
CRS grade 3–4	13 (14.5)	12 (24.0)	1 (2.5)	
Tocilizumab use (*n*, %)	61 (67.8)	36 (72.0)	25 (62.5)	0.337
Tocilizumab (doses)	1 (0–3)	1.5 (0–3.0)	1 (0–2)	0.128
ICANS incidence (*n*, %)	30 (33.3)	19 (38.0)	11 (27.5)	0.294
ICANS maximum grade (*n*, %)	0 (0–2)	0 (0–2)	1 (0–1)	0.257
ICANS ≥ 2 (*n*, %)	11 (12.2)	8 (16.0)	3 (7.5)	0.333
Serum IL-6 peak (pg/mL)	183.0 (34.0–1190.7)	257.9 (61.6–1679.5)	102.3 (24.7–677.2)	0.054
Serum IL-2R peak (UI/L)	4124.0 (2105.2–6153.2)	4323.5 (2227–6935)	3188.0 (1964–5578)	0.137
● CAR-T expansion (ddPCR assay, copies/microg gDNA) *n* = 70				
● Day 7	8683 (1616–20,831)	7147 (1101–21,532)	10618 (3391–20,375)	0.218
● Day 14	8395 (2543–17,962)	8520 (2500–19,850)	8270 (2789–14,800)	0.776
● Day 30	2387 (1644–13,191)	717.5 (200–4392)	749 (205–4451)	0.879
Neutropenia ≥ G3 (*n*, %)	54 (60.0)	36 (72)	18 (45)	0.084
Neutropenia phenotype (*n*, %)				
Quick recovery	47 (52.2)	22 (44.0)	25 (62.5)	
Intermittent recovery	16 (17.8)	9 (18.0)	7 (17.5)	**0.001**
Aplastic	20 (22.2)	18 (36.0)	2 (5.0)	
Early ICAHT > 2 (*n*, %)	26 (28.8)	22 (44.0)	4 (10.0)	**0.001**
Late ICAHT > 2 (*n* = 86) (n, %)	25 (27.7)	21 (42.0)	4 (10.0)	**0.005**
3-month ORR (*n* = 86) ** (*n*, %)	60 (69.7)	29 (58.8)	31 (77.5)	0.073

* Others include Rituximab + lenalidomide (*n* = 6), Inotuzumab ozogamicin (*n* = 4), polatuzumab vedotin + Rituximab (*n* = 2), polatuzumab vedotin (*n* = 1), ibrutinib (*n* = 1), and corticosteroid (*n* = 1). ** Patients who achieved a complete response (CR) or partial response (PR) were classified as ‘responders’, while all other patients were considered ‘non-responders’.

**Table 2 life-15-01419-t002:** Predictive factors of early and late transfusions after CAR-T-cell therapy in multivariate analysis.

	Early RBCs	Early Platelets	Late RBCs	Late Platelets
Transfusion support before CAR-T-cell therapy	**8.7** (2.1–36.9) *p* = 0.002	**8.3** (1.7–41.3) *p* = 0.009	**7.6** (1.4–41) *p* = 0.019	**8.1** (1.5–46.1) *p* = 0.017
High (≥2) CAR-HEMATOTOX score	**5.1** (1.4–18.7) *p* = 0.013	N.S.	N.S.	N.S.
Female sex	**6.3** (1.7–23.3) *p* = 0.005	N.S.	N.S.	N.S.
ECOG PS ≥ 2	N.S.	**4.9** (1.1–22.5) *p* = 0.041	N.S.	N.S.
No response to bridging therapy	**4.4** (1.2–16.0) *p* = 0.026	N.S.	N.S.	N.S.
Severe (grade > 2) ICANS	N.S.	**6.3** (1.1–38.6) *p* = 0.042	N.S.	N.S.
Severe (>2) early ICAHT	N.S.	N.S.	N.S.	**4.9** (1.1–22.2) *p* = 0.036

Associations are expressed as adjusted odds ratios and 95% confidence intervals. Values in bold signify statistical significance (*p* < 0.05). RBC, red blood cell. Early: <30 days post-infusion. Late: >30 days post-infusion. N.S, not significant.

**Table 3 life-15-01419-t003:** Survival outcomes, including median PFS and OS (with their respective 95% confidence intervals), for patients receiving early and late transfusions. The log-rank test was used to compare survival distributions between groups. Differences are expressed as hazard ratios (HRs, 95% confidence intervals).

	Median PFS (Months, 95% CI)				Median OS (Months, 95% CI)			
	**Transfused**	**Not Transfused**	**HR**	***p*-Value**	**Transfused**	**Not Transfused**	**HR**	** *p* ** **-Value**
Any RBCs	6.6 (3.5–19.3)	24.5 (14.2–28.6)	1.51 (0.84–1.71)	0.175	17.4 (12.0–23.6)	31.2 (28.6–40.0)	1.62 (0.77–3.41)	0.212
Any Platelets	5.8 (1.5–13.1)	24.5 (14.2–40.0)	**2.14** (1.11–4.50)	**0.013**	8.2 (3.7–20.7)	40.0 (28.6-NR)	**3.12** (1.17–8.33)	**0.001**
Early RBCs	6.6 (3.3–19.3)	24.5 (14.2–28.6)	1.50 (0.84–2.69)	0.177	17.4 (16.6–23.7)	31.2 (28.6–40.0)	1.51 (0.72–3.17)	0.282
Early Platelets	5.8 (1.4–13.1)	24.5 (6.6–40.0)	2.19 (1.05–5.04)	**0.016**	20.7 (2.9–20.7)	NR (28.6-NR)	**2.73** (1.11–8.29)	**0.011**
Late RBCs	6.6 (1.6–13.1)	19.2 (6.5–28.6)	1.15 (0.54–2.47)	0.706	7.6 (6.8–8.2)	40.0 (23.7-NR)	1.58 (0.59–4.19)	0.288
Late Platelets	6.0 (1.4–13.1)	16.8 (6.5–28.6)	1.36 (0.58–3.19)	0.421	7.1 (3.7–8.2)	40.0 (23.7-NR)	2.25 (0.74–6.78)	0.056

Values in bold signify statistical significance (*p* < 0.05). NR, not reached; RBC, red blood cell. Early: <30 days post-infusion. Late: >30 days post-infusion.

## Data Availability

The data that support the findings of this study are available from the corresponding author upon reasonable request.

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
