# Peer review of "Deciphering the Complex Intertwining Between Cytopenia and Transfusion Needs After CAR-T-Cell Therapy for B-Cell Malignancies"

_life, 2025, doi:10.3390/life15091419_

Round 1

Reviewer 1 Report

Comments and Suggestions for Authors OVERALL COMMENTS This study bears a high degree of similarity to the study by Vic et. Al. (Reference 8 of this paper) on transfusion therapy post CART for large B cell lymphoma. Both studies examined very similar baseline parameters, their impact on transfusion needs, and the correlation with  long term outcomes.  The analyses performed are very similar and the data are presented in very similar formats in the two studies. However, as acknowledged by the authors  this study is much smaller  (n=90 vs 691) and more heterogenous (80% DLBCL, 20% other), it is also single center rather than multi-center. Despite these limitations (and other specific issues described below), I am in favor of publishing this study provided satisfactory revision because:
  1. Data on transfusion support after CART, and potential prognostic value on long term outcomes are still very limited and needed, as CART becomes more widely available
  2. This study was based in a different country (Italy vs. France) and some regional differences in transfusion or CART clinical practices might be expected. The authors may want to discuss these differences if they are pertinent. It is reassuring that the results are largely in agreement of the earlier larger study.
  3. While some parameters were not included (e.g. previous HSCT history), the investigators included new parameters in their study, (BM cellularity, types response to CART, gender etc). Of note, the authors also used a variety of parameters to investigate the presence of longitudinal neutropenia, and provide data on CART-expansion, serum cytokine levels, BM infiltration of disease  , all of which were cited as shortcomings of the Vic study by the accompanying editorial(Alhomoud, Blood Advances, 2024; 8:1570-572). The authors made conscious and laudable efforts to address these issues and to further highlight the transfusion burdens associated with CART and provide  insight on its pathophysiology and clinical impact.
SPECIFIC Comments;
  1. The author should consider a more focused/detailed comparative discussion on the study by Vic et al. and highlight new findings in the current study or differences between the two studies, relative limitations and advantages.
  2. Figure 3 is extremely difficult to see and many labels are not legible in the current PDF reviewed by me. Please enlarge and use higher resolution. 
  3.  A "Table 3" is referred to in the text along with figure 3, presumably showing the relationship between transfusion and survival outcomes, NRM etc. However, I cannot find such table in the document . This makes it difficult to evaluate the  key conclusions and discussions regarding impact on outcomes.
  4. Will figures all be in color? If not, many of the figures would be difficult to read in black and white. Please consider using dashed vs. solid lines.

  1. Some platelets were given as pool platelets products. What is the pool size? Is there variability in the size of the pool? Is one pool always treated as equivalent to one apheresis unit in the analyses?
  2. What addtional value does the inclusion of TRFS as an endpoint add? (TRFS is defined as survival and the last follow-up without transfusion or relapse, new treatment or death. Please elucidate or consider omitting.
  3. In tables, values in parentheses representing percentages, vs.  ranges should be labeled or differentiated as such clearly. Units should be provided consistently (e.g. ddPRC assay value, copies/microliter)
  4. Please provide specific definitions (or references) for neutrophil phenotype, quick, intermittent recovery vs aplastic
  5. Please provide the range of the numbers of transfused units in addition to median and IQR. The IQR is fairly wide and one wonders for future studies, whether  further quantitative data analyses ( in addition to a qualitative yes vs no regarding transfusions) such as the amount of transfusions required, or time to transfusion independence could be useful parameters to examine
  6. I have trouble understanding the paragraph on lines 326--330
  7. Reference section still contains instruction to authors, which should be removed

Author Response

Thank you very much for taking the time to review this manuscript. Please find the detailed responses below and the corresponding revisions/corrections highlighted in the re-submitted files

OVERALL COMMENTS This study bears a high degree of similarity to the study by Vic et. Al. (Reference 8 of this paper) on transfusion therapy post CART for large B cell lymphoma. Both studies examined very similar baseline parameters, their impact on transfusion needs, and the correlation with long term outcomes.  The analyses performed are very similar and the data are presented in very similar formats in the two studies. However, as acknowledged by the authors this study is much smaller (n=90 vs 691) and more heterogenous (80% DLBCL, 20% other), it is also single center rather than multi-center.

Despite these limitations (and other specific issues described below), I am in favor of publishing this study provided satisfactory revision because:

  1. Data on transfusion support after CART, and potential prognostic value on long term outcomes are still very limited and needed, as CART becomes more widely available
  2. This study was based in a different country (Italy vs. France) and some regional differences in transfusion or CART clinical practices might be expected. The authors may want to discuss these differences if they are pertinent. It is reassuring that the results are largely in agreement of the earlier larger study.
  3. While some parameters were not included (e.g. previous HSCT history), the investigators included new parameters in their study, (BM cellularity, types response to CART, gender etc). Of note, the authors also used a variety of parameters to investigate the presence of longitudinal neutropenia, and provide data on CART-expansion, serum cytokine levels, BM infiltration of disease, all of which were cited as shortcomings of the Vic study by the accompanying editorial (Alhomoud, Blood Advances, 2024; 8:1570-572). The authors made conscious and laudable efforts to address these issues and to further highlight the transfusion burdens associated with CART and provide insight on its pathophysiology and clinical impact.

Re: We sincerely thank the reviewer for these constructive remarks.

SPECIFIC Comments;

  1. The author should consider a more focused/detailed comparative discussion on the study by Vic et al. and highlight new findings in the current study or differences between the two studies, relative limitations and advantages.

Re: we thank the Reviewer for this valuable consideration. In the Discussion section, we expanded the comparative analysis with the study by Vic et al.  according to the suggestions provided. Specifically, we emphasized both the differences and the similarities between the two studies, outlining relative limitations and strengths (line 400-412)

  1. Figure 3 is extremely difficult to see and many labels are not legible in the current PDF reviewed by me. Please enlarge and use higher resolution. 

Re: We sincerely apologize for this inconvenience. The figure has now been revised and replaced with an enlarged, higher-resolution version to ensure that all labels and details are clearly legible.

  1.  A "Table 3" is referred to in the text along with figure 3, presumably showing the relationship between transfusion and survival outcomes, NRM etc. However, I cannot find such table in the document. This makes it difficult to evaluate the key conclusions and discussions regarding impact on outcomes.

Re: We sincerely apologize for this inconvenience. The missing Table 3 has been added to the revised manuscript.

  1. Will figures all be in color? If not, many of the figures would be difficult to read in black and white. Please consider using dashed vs. solid lines.

Re:  We thank the reviewers for this observation. All figures will be in color in the final version.

  1. Some platelets were given as pool platelets products. What is the pool size? Is there variability in the size of the pool? Is one pool always treated as equivalent to one apheresis unit in the analyses?

Re:  We thank the reviewer for raising this important point. All pooled platelet concentrates transfused in our study were prepared in accordance with the Guide to the preparation, use and quality assurance of blood components from the European Directorate for the Quality of Medicines & HealthCare (EDQM). Each pool consisted of a leukocyte-depleted, pathogen-inactivated platelet component derived from 4 to 6 fresh whole blood donations. Both pooled platelet products and apheresis platelets contained a minimum of 2 × 10¹¹ platelets per unit. At the time of transfusion, the type of platelet component (pooled vs. apheresis) was randomly assigned according to blood bank availability. For the purposes of the analyses, one pooled platelet product was considered equivalent to one apheresis platelet unit. This information has now been clarified in the revised Methods section (line 97-100).

  1. What additional value does the inclusion of TRFS as an endpoint add? (TRFS is defined as survival and the last follow-up without transfusion or relapse, new treatment or death. Please elucidate or consider omitting.

Re:  We thank the reviewer for this valuable comment. We included TRFS as a parameter to capture how early a patient required transfusion after CAR-T therapy, going beyond the simple categorical yes/no assessment. In our cohort, a shorter time to first transfusion was associated with a higher total transfusion burden, the development of early and late ICAHT, and lower response rates. These findings highlight the potential utility of TRFS as a simple early marker to guide closer monitoring and supportive care strategies in most vulnerable CAR-T patients.

  1. In tables, values in parentheses representing percentages, vs.  ranges should be labelled or differentiated as such clearly. Units should be provided consistently (e.g. ddPRC assay value, copies/microliter).

Re: We thank the reviewer for this useful observation. All tables have been revised to ensure consistency and clarity. Values in parentheses are now explicitly labelled as either percentages or ranges, to avoid any ambiguity. Units of measurement have also been provided consistently across all variables.

  1. Please provide specific definitions (or references) for neutrophil phenotype, quick, intermittent recovery vs aplastic.

Re: We thank the Reviewer for the suggestion. In the revised version, we have provided the definitions of the neutrophil phenotypes (quick, intermittent recovery, and aplastic) and included an appropriate reference in the Methods section (line 89-93).

  1. Please provide the range of the numbers of transfused units in addition to median and IQR. The IQR is fairly wide and one wonders for future studies, whether further quantitative data analyses (in addition to a qualitative yes vs no regarding transfusions) such as the amount of transfusions required, or time to transfusion independence could be useful parameters to examine).

Re: We thank the Reviewer for this insightful comment. In the revised manuscript, we have now added the range of transfused units in addition to the median and IQR, as suggested (line 181-189).  We fully agree that future studies could benefit from more detailed quantitative analyses (e.g., cumulative number of transfusions), beyond the dichotomous yes/no assessment, and we have acknowledged this point in the Discussion section (line 392-394).

  1. I have trouble understanding the paragraph on lines 326—330

Re: We thank the Reviewer for pointing this out and we apologize for the lack of clarity. We have revised the paragraph to improve readability and better convey the intended message (line 369-375).

  1. Reference section still contains instruction to authors, which should be removed

Re: We sincerely apologize for this inconvenience. Instruction to authors were removed in the revised version.

Reviewer 2 Report

Comments and Suggestions for Authors

In the manuscript titled "Deciphering the Complex Intertwining Between Cytopenia and Transfusion Needs After CAR T-Cell Therapy for B-Cell Malignancies," a team led by Pellegrino and Teofili investigates the predictive factors associated with transfusion requirements in patients receiving anti-CD19 CAR-T cell therapy for B-cell malignancies.

Overall, the authors present interesting points in a well-structured manner. However, there are a few weaknesses that need to be addressed before the manuscript can be accepted.

  1. This study is quite similar to another paper published in 2024 (PMID: 38181767). The authors should provide more discussion to highlight their novelty or differences compared to that study.
  2. Since transfusion of blood components is unavoidable in patients treated with CAR-T cells, additional discussion on management recommendations is necessary.
  3. In the methods section, the authors may consider providing more details about the Sankey diagram.
  4. Additionally, Figure 3 is of low resolution, and the numbers are too small to read. 

Author Response

REVIEWER 2

Thank you very much for taking the time to review this manuscript. Please find the detailed responses below and the corresponding revisions/corrections highlighted in the re-submitted files

In the manuscript titled "Deciphering the Complex Intertwining Between Cytopenia and Transfusion Needs After CAR T-Cell Therapy for B-Cell Malignancies," a team led by Pellegrino and Teofili investigates the predictive factors associated with transfusion requirements in patients receiving anti-CD19 CAR-T cell therapy for B-cell malignancies.

Overall, the authors present interesting points in a well-structured manner. However, there are a few weaknesses that need to be addressed before the manuscript can be accepted.

  1. This study is quite similar to another paper published in 2024 (PMID: 38181767). The authors should provide more discussion to highlight their novelty or differences compared to that study.

 Re: we thank the Reviewer for this valuable suggestion. In the Discussion section, we expanded the comparative analysis with the study by Vic et al. Specifically, we emphasized both the differences and the similarities between the two studies, outlining relative limitations and strengths (line 400-412)

  1. Since transfusion of blood components is unavoidable in patients treated with CAR-T cells, additional discussion on management recommendations is necessary.

Re: We thank the Reviewer for this suggestion. In the Introduction, we have expanded the discussion of current practice regarding transfusion management in CAR T-cell patients, including transfusion thresholds and the requirement for irradiation (Lines 54–62).

  1. In the methods section, the authors may consider providing more details about the Sankey diagram.

Re: we thank the Reviewer for this suggestion. In the revised methods section, we have added more details regarding the Sankey diagram, specifying the time intervals analyzed (the three months before and after CAR-T), the transfused components (RBCs and platelets), and the rationale for using this visualization to capture transitions in transfusion status.

  1. Additionally, Figure 3 is of low resolution, and the numbers are too small to read. 

Re: We sincerely apologize for this inconvenience. The figure has now been revised and replaced with an enlarged, higher-resolution version to ensure that all labels and details are clearly legible.

Reviewer 3 Report

Comments and Suggestions for Authors

This is a well-conducted and clinically relevant retrospective study exploring transfusion needs and cytopenia after CAR T-cell therapy. The topic is timely, and the data are presented in a mostly clear and interpretable manner. That said, several aspects would benefit from clarification or further elaboration:

Major Points:

1. The study observes a clear association between platelet transfusion and poorer survival outcomes. However, this association should not be overinterpreted as causation. It’s important to acknowledge and discuss potential confounding factors more explicitly—especially since patients receiving transfusions were more likely to have poor baseline characteristics (e.g., higher ECOG, worse BM status). If possible, consider a multivariate Cox regression adjusting for major prognostic variables to strengthen this point.

2. The authors state that transfusion thresholds were consistent throughout the study. Please clarify whether any changes in institutional guidelines or clinical practice occurred during the long study period (2019–2024). This would help readers understand the comparability of transfusion data across time.

3. While EPO use is briefly mentioned, data on G-CSF (which is more commonly used post-CAR-T) are lacking. Please clarify whether G-CSF was used, and if so, how it may have influenced transfusion need or ICAHT recovery.

4. Reduced BM cellularity was significantly associated with transfusion need. Could the authors elaborate on how cellularity was assessed? Was this based on pathology reports, a standardized scoring system, or subjective interpretation?

5. The discussion briefly mentions the possibility of transfusion-related immune modulation. This is an interesting point that could be expanded slightly, particularly in the context of cellular therapies.

Minor Points:

1. There are minor language issues throughout (e.g., “receveid” → “received”; “cart” → “CAR-T” should be consistent). A thorough copyediting pass would improve readability.

2. Please ensure all abbreviations are defined on first use in the main text. While an abbreviation list is provided, this step improves readability.

Author Response

REVIEWER 3

Thank you very much for taking the time to review this manuscript. Please find the detailed responses below and the corresponding revisions/corrections highlighted in the re-submitted files

This is a well-conducted and clinically relevant retrospective study exploring transfusion needs and cytopenia after CAR T-cell therapy. The topic is timely, and the data are presented in a mostly clear and interpretable manner. That said, several aspects would benefit from clarification or further elaboration:

Major Points:

  1. The study observes a clear association between platelet transfusion and poorer survival outcomes. However, this association should not be overinterpreted as causation. It’s important to acknowledge and discuss potential confounding factors more explicitly—especially since patients receiving transfusions were more likely to have poor baseline characteristics (e.g., higher ECOG, worse BM status). If possible, consider a multivariate Cox regression adjusting for major prognostic variables to strengthen this point.

Re: We are grateful to the reviewer for bringing up this relevant point. We fully agree that association does not mean causation, and we further stressed this aspect in the discussion. Moreover, we performed a multivariate Cox regression analysis to adjust for variables with conceivable effect on survival. The corresponding text has been updated in the Methods (line 134-136) and Results sections (line 283-286).

  1. The authors state that transfusion thresholds were consistent throughout the study. Please clarify whether any changes in institutional guidelines or clinical practice occurred during the long study period (2019–2024). This would help readers understand the comparability of transfusion data across time.

Re: We thank the reviewer for this comment. We confirm that no changes in institutional guidelines or clinical practice regarding transfusion thresholds occurred during the study period. The criteria for transfusion remained stable, ensuring full comparability of the data across different time intervals. We better clarified this aspect in the methods section (line 94-95)

  1. While EPO use is briefly mentioned, data on G-CSF (which is more commonly used post-CAR-T) are lacking. Please clarify whether G-CSF was used, and if so, how it may have influenced transfusion need or ICAHT recovery.

Re: We thank the reviewer for raising this important point. G-CSF was not used routinely as pre-emptive or prophylactic therapy in any patient of our cohort. In line with our institutional practice, its administration was limited to the outpatient setting, and only a small fraction of patients received it for a short duration of time without producing stable increases in neutrophil counts. Given the very small number of patients treated, no statistical conclusions could be drawn regarding its effect. However, we do not expect this limited use to have substantially influenced ICAHT recovery or transfusion requirements.

  1. Reduced BM cellularity was significantly associated with transfusion need. Could the authors elaborate on how cellularity was assessed? Was this based on pathology reports, a standardized scoring system, or subjective interpretation?

Re: We thank the reviewer for this comment. Bone marrow cellularity is routinely assessed in bone marrow biopsies by our pathologists, who quantitatively measures the hematopoietic area and compares it with age-specific reference values reported in the literature (Thiele et al, 2005). Cellularity is then classified as hypo-, normo-, or hypercellular accordingly. This methodology has been clarified in the revised Methods section (line 79-80).

  1. The discussion briefly mentions the possibility of transfusion-related immune modulation. This is an interesting point that could be expanded slightly, particularly in the context of cellular therapies.

Re:  We thank the reviewer for this suggestion. We have briefly expanded the discussion to better describe transfusion-related immune modulation and to note that it may plausibly influence the function and persistence of CAR‑T cells, although its impact in this setting remains poorly defined (line 325-333).

Minor Points:

  1. There are minor language issues throughout (e.g., “receveid” → “received”; “cart” → “CAR-T” should be consistent). A thorough copyediting pass would improve readability.

Re: We sincerely apologize for this inconvenience. We have carefully proofread the manuscript to correct typographical and language errors, ensuring consistency in terminology

  1. Please ensure all abbreviations are defined on first use in the main text. While an abbreviation list is provided, this step improves readability.

Re: We thank the Reviewer for this comment. We have modified the manuscript accordingly.

Round 2

Reviewer 2 Report

Comments and Suggestions for Authors

The author addressed all the comments, and the manuscript was much improved. No further comments.